# Effect of a Multidimensional Physical Activity Intervention on Body Mass Index, Skinfolds and Fitness in South African Children: Results from a Cluster-Randomised Controlled Trial

**DOI:** 10.3390/ijerph16020232

**Published:** 2019-01-15

**Authors:** Ivan Müller, Christian Schindler, Larissa Adams, Katharina Endes, Stefanie Gall, Markus Gerber, Nan S. N. Htun, Siphesihle Nqweniso, Nandi Joubert, Nicole Probst-Hensch, Rosa du Randt, Harald Seelig, Danielle Smith, Peter Steinmann, Jürg Utzinger, Peiling Yap, Cheryl Walter, Uwe Pühse

**Affiliations:** 1Swiss Tropical and Public Health Institute, P.O. Box, CH-4002 Basel, Switzerland; christian.schindler@swisstph.ch (C.S.); shwenwetun@gmail.com (N.S.N.H.); nicole.probst@swisstph.ch (N.P.-H.); peter.steinmann@swisstph.ch (P.S.); juerg.utzinger@swisstph.ch (J.U.); peiling.yap@gmail.com (P.Y.); 2University of Basel, P.O. Box, CH-4003 Basel, Switzerland; 3Department of Sport, Exercise and Health, University of Basel, Birsstrasse 320 B, CH-4052 Basel, Switzerland; katharina.endes@unibas.ch (K.E.); stefanie.gall@unibas.ch (S.G.); markus.gerber@unibas.ch (M.G.); harald.seelig@unibas.ch (H.S.); uwe.puehse@unibas.ch (U.P.); 4Department of Human Movement Science, South Campus, P.O. Box 77000, Nelson Mandela University, Port Elizabeth 6031, South Africa; larissa.adams13@gmail.com (L.A.); felicitas.nqweniso@mandela.ac.za (S.N.); nandi.joubert@mandela.ac.za (N.J.); rosa.durandt@mandela.ac.za (R.d.R.); danielle.smith@mandela.ac.za (D.S.); cheryl.walter@mandela.ac.za (C.W.); 5Institute of Infectious Disease and Epidemiology, Tan Tock Seng Hospital, 144 Moulmein Road, Singapore 308089, Singapore

**Keywords:** body mass index, cardiorespiratory fitness, intestinal protozoa, physical activity programme, school-aged children, soil-transmitted helminths, South Africa

## Abstract

Obesity-related conditions impose a considerable and growing burden on low- and middle-income countries, including South Africa. We aimed to assess the effect of twice a 10-week multidimensional, school-based physical activity intervention on children’s health in Port Elizabeth, South Africa. A cluster-randomised controlled trial was implemented from February 2015 to May 2016 in grade 4 classes in eight disadvantaged primary schools. Interventions consisted of physical education lessons, moving-to-music classes, in-class activity breaks and school infrastructure enhancement to promote physical activity. Primary outcomes included cardiorespiratory fitness, body mass index (BMI) and skinfold thickness. Explanatory variables were socioeconomic status, self-reported physical activity, stunting, anaemia and parasite infections. Complete data were available from 746 children. A significantly lower increase in the mean BMI Z-score (estimate of difference in mean change: −0.17; 95% confidence interval (CI): −0.24 to −0.09; *p* < 0.001) and reduced increase in the mean skinfold thickness (difference in mean change: −1.06; 95% CI: −1.83 to −0.29; *p* = 0.007) was observed in intervention schools. No significant group difference occurred in the mean change of cardiorespiratory fitness (*p* > 0.05). These findings show that a multidimensional, school-based physical activity intervention can reduce the increase in specific cardiovascular risk factors. However, a longer and more intensive intervention might be necessary to improve cardiorespiratory fitness.

## 1. Introduction

Non-communicable diseases (NCDs) and obesity-related conditions, such as diabetes and cardiovascular diseases, impose a considerable and rapidly growing burden on low- and middle-income countries (LMICs) [1]. Physical inactivity and unhealthy diet, particularly low vegetable and fruit intake and excess salt and sugar consumption, have emerged as new leading risk factors, accounting for 10% of the global burden of disease [2]. In 2010, for the first time, overweight replaced under-nutrition as a risk factor, as determined in the Global Burden of Disease (GBD) study [2]. Indeed, overweight is a rapidly growing epidemic affecting all socioeconomic strata and ethnicities [3]. Factors that contribute to the increase in average body fat are excessive energy intake governed by fast food consumption, and a decrease in energy expenditure through physical inactivity, partially explained by sedentary behaviour (e.g., television viewing and personal transport by automobile) [4,5]. A meta-analysis by Guerra and colleagues with data from 11 randomised trials suggests that, regardless of the potential benefits of physical activity in the school environment, school-based physical activity interventions did not have any significant effects on body mass index (BMI) [6], while another non-randomised trial by Li et al. reported favourable effects [7].

Representative surveys revealed that the South African population has moved towards a disease profile similar to Western countries, where a considerable proportion of preschool- and school-aged children are overweight or obese, and increasing proportions of deaths among adults are attributed to chronic diseases of lifestyle [8,9]. South Africa’s 2018 Report Card on Physical Activity for Children and Youth highlights the current concerns for health and wellbeing of children and youth related to declining physical activity levels [10]. Moreover, South Africans consume about three times the global average of soft drinks, and intake of fast food is reported at least three times a week by more than two thirds of adolescents [10]. Meanwhile, socioeconomically deprived communities with a high burden of infectious diseases persist in South Africa [11,12,13].

In the paper presented here, we examined whether participation in the ‘Disease, Activity and Schoolchildren’s Health’ (DASH) multidimensional physical activity intervention programme would improve children’s cardiorespiratory fitness and counteract an excess increase of BMI and skinfold thickness. We took into account baseline adjustments for potential sociodemographic confounders and soil-transmitted helminth infection status. The overarching purpose of the DASH study was to investigate the dual disease burden (i.e., NCDs and infectious diseases) among children in primary schools in disadvantaged neighbourhoods [14].

## 2. Methods

### 2.1. Study Area and Population

The study was carried out in Port Elizabeth in the Eastern Cape province of South Africa (geographical coordinates: 34°07′54″ to 33°57′29″ S latitude and 25°36′00″ to 25°55′49″ E longitude). Recruitment of schools commenced in September 2014 and two 10-week multidimensional physical activity interventions were implemented in July-September 2015 and February-April 2016. Overall, 103 quintile 3 primary schools were eligible for participation. South Africa’s public schools are classified into five groups, with quintile five standing for the least poor and quintile one standing for the poorest. The quintiles are determined through the national poverty table, prepared by the treasury. Areas are being ranked on the basis of income levels, dependency ratios and literacy rates in the area. The quintile ranking of a school determines the no-fee status of the school and also the amount of money that a school receives, with the poorest schools receiving the greatest per-learner allocation [15].

From the 103 quintile 3 schools, 25 schools expressed an interest, as documented in a response letter. Those 25 schools were invited to an information sharing meeting that was attended by 15 schools. Among the 15 schools, seven did not satisfy the chief criterion of having at least 100 learners in grade 4, and hence, were excluded. Eight schools were selected based on (i) sufficiently large grade 4 classes (*n* > 100 children); (ii) geographical location; (iii) representation of the various target communities; and (iv) commitment to support the project activities (Figure 1). In South Africa, physical education is generally neglected in disadvantaged schools due to resource limitations and priority given to academic subjects, although two weekly lessons are officially included in the curriculum.

The study population consisted of coloured children (mixed race ancestry), usually Afrikaans speaking, and black African children, mainly Xhosa speaking. Children’s age ranged between 9 and 14 years with a mean of 11.2 years (standard deviation 0.9 years) [14]. The following inclusion criteria were employed: (i) willingness to participate; (ii) written informed consent by a parent/guardian; (iii) no participation in other clinical trials during the study period; and (iv) not suffering from medical conditions preventing participation in a maximum exercise test, as determined by qualified medical personnel.

### 2.2. Study Design and Randomisation

Enrolment of schools was conducted by the research team. The DASH study was designed as a cluster-randomised controlled trial. In order not to contaminate intervention effects, schools rather than classes were randomised (Appendix A). Generating the allocation sequence by a simple randomisation of the schools was carried out by the research team on the basis of a computer-generated random number list. Among the eight available schools, three different interventions were implemented in four different combinations. One of the interventions, which is the primary focus of the current paper, involved additional physical activity lessons (PA), while the other two consisted in a health and hygiene education (HE) and a nutrition education (NU) programme, respectively. First, four schools not to get any intervention were randomly selected, and then each of the four remaining schools was randomly allocated to one of the following intervention combinations: (a) PA + HE + NU; (b) PA + HE; (c) PA alone; and (d) HE + NU [14,16] (Appendix A). The PA intervention was thus carried out in three of the eight schools. No financial incentives were provided; neither to the school authorities, nor to the participating children.

### 2.3. Interventions

We developed a multidimensional physical activity intervention in collaboration with education authorities, teachers and students from the participating schools. As shown in Figure 2, the multidimensional physical activity intervention programme consisted of four components: (i) two 40 min physical education lessons per week; (ii) one weekly 40 min moving-to-music lesson; (iii) regular in-class physical activity breaks incorporated into the main school curriculum; and (iv) enhancement of the school environment to be more physical activity friendly (e.g., installation of activity stations and a variety of painted games).

The intervention components were applied in schools during official hours, and were taught in collaboration with the life orientation teachers. The dancing lessons were led by students from the Nelson Mandela University. The control group continued to follow their usual school curriculum. At the beginning of the study, we carried out project information sharing sessions with principals, teachers, parents and school governing bodies. Workshops with the life orientation teachers and class teachers were organised to discuss the materials and information provided during the intervention. Physical education lessons were given twice per week covering two 10-week periods. From the first 10-week intervention block to the second, the research team optimised the intervention components to render them more efficient with an increased intensity. All the lessons adhered to the requirements of the South African Curriculum and Assessment Policy Statement (CAPS). Teachers designated to provide the physical education were assisted by a trained physical education coach for one of the two weekly lessons, while the teachers thereafter taught the subsequent lesson on their own. A physical education lesson lasted 40 min, starting with a 5 min warm-up and concluding with a cool-down towards the end of the lesson. The physical education classes were taught outside on either grass or cemented areas and most children wore light sports clothing. Sports equipment for the lessons (e.g., bean bags, colour bands, skipping ropes, cones and diverse balls) was donated to the schools. In addition to the physical activity intervention, two supplementary programmes were conducted in selected schools. The first one was a health and hygiene education programme to increase children’s awareness for communicable diseases and the second one a nutrition education and supplementation programme to contribute to the awareness of healthy diet, as described in the study protocol published elsewhere [14] and summarised in the Appendix A of the current piece (Appendix A).

### 2.4. Ethics Statement

All subjects gave their informed consent for inclusion before they participated in the study. The study was conducted in accordance with the Declaration of Helsinki and was cleared by the ethics committees of Northwest and Central Switzerland (reference no. 2014-179), the Nelson Mandela University (study number H14-HEA-HMS-002), the Eastern Cape Department of Education and the Eastern Cape Department of Health. The study is registered at ISRCTN registry under controlled-trials.com (identifier: ISRCTN68411960). There were no injuries or other adverse events during the physical activity lessons. Soil-transmitted helminth infections were managed according to guidelines put forth by the World Health Organization (WHO) and national treatment recommendations.

### 2.5. Procedures

Baseline and endline measurements took place at eight disadvantaged quintile 3 primary schools before and after the implementation of the multidimensional physical activity intervention. Primary outcome measures included anthropometric indicators (i.e., height, weight and skinfolds; the latter including triceps and subscapular) and cardiorespiratory fitness. Secondary outcomes were socioeconomic status (SES), self-reported physical activity, haemoglobin (Hb) and infection with soil-transmitted helminths and intestinal protozoa.

Children’s body weight was measured once to the nearest 0.1 kg (Micro T7E electronic platform scale, Optima Electronics; George, South Africa). Height was assessed to the nearest 0.1 cm with a Seca stadiometer (Surgical SA; Johannesburg, South Africa). Height and BMI (defined as weight [kg]/height [m]^2^) were standardised according to WHO guidelines, resulting in HAZ (height for sex and age) and BMIZ (BMI for sex and age) scores [17]. Stunting was defined as sex-adjusted height-for-age Z-score (HAZ) < −2, overweight as BMIZ score of +1 to +2, and obesity as BMIZ score of > +2. Skinfold-thickness was measured three times at triceps and subscapular positions, using a Harpenden skinfold caliper [18,19].

Cardiorespiratory fitness was assessed using the 20 m shuttle run test, adhering to a standard test protocol [20,21]. Most of the schoolchildren wore school or street shoes, while some ran barefoot. In brief, when a child failed to follow the pace for two consecutive intervals, the child was asked to stop running. As the final score, we noted the number of fully completed 20 m laps and converted the results in VO_2_ max values, according to a standard protocol [21].

We measured Hb concentration once to the nearest 0.01 g dL^−1^ with a HemoCue^®^ Hb 301 system (HemoCue^®^AB; Ängelholm, Sweden). After swabbing the child’s fingertip with alcohol, a nurse or qualified field worker pricked the fingertip with a safety lancet and squeezed gently to obtain two drops of blood. The first drop was wiped away with the alcohol swab and the second drop was collected with a microcuvette. Anaemia was defined as an Hb concentration ≤11.4 g dL^−1^ [22].

The parasitological work-up is detailed in the study protocol [14]. In brief, the Kato-Katz technique was used on the stool samples to identify and count the number of soil-transmitted helminth eggs that were reported for each species separately. Additionally, a Crypto-Giardia Duo-Strip^®^ rapid diagnostic test (RDT) (CORIS, BioConcept; Gembloux, Belgium) was performed for the detection of *Cryptosporidium* spp. and *Giardia intestinalis* [23]. Deworming was offered to infected children or the entire school, depending on school prevalence according to WHO and national treatment guidelines [24]. The SES index is based on household asset ownership and housing characteristics, determined by a questionnaire completed by the participants, which also examined self-reported physical activity (Health Behaviour in School-aged Children; HBSC) (Appendix A) [25]. The detailed test description is provided in the study protocol [14].

### 2.6. Statistical Analysis

Details of the sample size calculation have been described elsewhere [14]. In brief, sample size was based on achieving sufficient precision in assessing the prevalence of soil-transmitted helminth infections at baseline, taking into account clustering within schools and classes as well as loss to follow-up, amounting to a total of 1200 participants. We used linear mixed models with random intercepts of schools and classes to analyse changes in quantitative outcomes (cardiorespiratory fitness, BMIZ, skinfolds and self-reported physical activity) from baseline to follow-up. The effects of the three interventions, physical activity lessons, health and hygiene education and nutritional education, on these changes were simultaneously estimated using separate indicator variables for the three types of intervention. These analyses were adjusted for sex, age, HAZ, SES index, Hb, intestinal protozoa (*Cryptosporidium* spp. and/or *G. intestinalis*) and soil-transmitted helminth infection (*Ascaris lumbricoides* and *Trichuris trichiura*). To capture potential ceiling effects, we also ran models adjusting for the baseline values of the respective outcome variable.

In addition, we assessed intervention effects on the occurrence of symptoms or infections at follow-up (i) among children having shown the respective condition already at baseline and (ii) among children who did not show the respective condition before. Mixed logistic regression models were used to analyse intervention effects on the occurrence of the respective condition at follow-up in the respective subsample. They included the same random intercepts and covariates as the previously described linear mixed models.

To analyse intervention effects on the prevalence of symptoms or infections, we used mixed logistic regression models including the occurrence of the respective condition at baseline and follow-up as repeated outcomes. These models were adjusted for the same baseline covariates as the previous models and included a random intercept for each child in addition to the random intercepts of schools and classes. To distinguish baseline and follow-up observations, an indicator variable for period with values 0 for baseline and 1 for follow-up was introduced. Intervention effects on the prevalence of a given condition were estimated using interaction terms between the intervention indicator variables and this period variable. In accordance with the focus of the present paper, only the estimated effects of the physical activity intervention are reported.

Furthermore, differences at baseline between the intervention and control group with regard to primary outcome measures, such as obesity, skinfolds and cardiorespiratory fitness, were investigated. We also examined potential heterogeneity of the intervention effects across subgroups (e.g., males vs. females; lower vs. higher SES). This was done by stratification into two categories of the respective variable (using a median split in case of quantitative characteristics) [26]. An additional analysis, excluding one of the schools, was conducted according to a factorial design (Appendix A). Statistical analyses were performed using STATA version 13.0 (STATA Corp.; College Station, TX, USA). Statistical significance was defined as *p* < 0.05.

## 3. Results

Of the 1009 grade 4 schoolchildren with written informed consent from their parents/guardians, complete baseline data were available from 746 children (Figure 1). Reasons for exclusion were withdrawal, absence during baseline data collection and incomplete baseline data (e.g., reported health problems precluding participation in the cardiorespiratory fitness test). All subsequent analyses pertained to the 746 children (372 girls, 49.9%; and 374 boys, 50.1%; mean age 10.0 years). 519 children presented a full data record after endline, of which 264 children participated in the physical activity intervention. No adverse events occurred during the physical activity intervention.

Table 1 summarises the baseline characteristics of the study participants, stratified by schools with physical activity intervention and schools without physical activity intervention. No significant differences in primary outcome measures, such as obesity, skinfolds and cardiorespiratory fitness at baseline were detected, when comparing schools with and without physical activity intervention (all *p* > 0.05).

Cardiorespiratory fitness and body composition results are presented in Table 2, along with adjusted estimates of intervention effects defined as difference in mean changes. We observed a significantly lower increase in mean BMIZ (estimated difference in mean change: −0.17; 95% confidence interval (CI): −0.24 to −0.09; *p* < 0.001) and reduced increase in the mean thickness of skinfolds (difference in mean change: −1.06; 95% CI: −1.83 to −0.29; *p* = 0.007) from baseline to endline when comparing children from the intervention group with their peers in the control group (intervention effects on primary outcome measures from baseline to midline and from midline to endline can be found in the Appendix A, respectively).

Mean changes in 20 m shuttle run test results and VO_2_ max were comparable between the two groups with a positive but statistically non-significant tendency for the intervention group. Self-reported physical activity remained stable in the intervention group but increased in the control group, resulting in a non-significant difference in mean change of −1.08 (95% CI: −2.36 to 0.18; *p* = 0.09). Intra-class correlations were all <0.05, showing a low level of unexplained variability between schools and classes. In the intervention group, the frequency of overweight children slightly declined from 12.7% to 11.5%, while obesity slightly increased from 5.0% to 6.5%. In the control group, both the prevalence of overweight and obese children increased (overweight from 14.4% to 17.2% and obesity from 5.4% to 8.5%). Stratified estimates of intervention effects on primary outcome measures are presented in Figure 3.

In particular, differential intervention effects on skinfold thickness were observed according to sex (with a higher benefit among girls), age (with higher benefit among younger children) and to SES (with higher SES being advantageous) as well as between anaemic and non-anaemic children in which the intervention had opposite effects.

Table 3 summarises the baseline and endline measurements of binary outcome variables (all secondary outcomes) affecting cardiorespiratory fitness or body composition, along with a description of the intervention effects on the re-occurrence (proportion of lasting cases), new occurrence (proportion of new cases) and prevalence (proportion of cases) of the respective conditions, as expressed by odds ratios (ORs). For none of the binary outcomes considered, the change in prevalence between baseline and endline, the new occurrence or the re-occurrence at follow-up was significantly associated with the intervention. At baseline, the prevalences of stunting, anaemia and infection with either soil-transmitted helminths or intestinal protozoa were 6.3%, 6.4%, 20.9% and 7.1% higher in the intervention (*n* = 300) than in the control group (*n* = 446), while after 16 months, the same indicators were 5%, 0.9%, 27.3% and 10.3% higher, respectively.

## 4. Discussion

A two-times 10-week multidimensional physical activity intervention contributed to a lower increase of BMI and thickness of skinfolds in schoolchildren from disadvantaged neighbourhoods in Port Elizabeth, South Africa. However, contrary to our hypothesis, no detectable effects were observed on cardiorespiratory fitness at the 16-month endline survey. Result modification for skinfold thickness and BMI by sex suggests that gender-specific interventions may be necessary. Although girls benefited more from the intervention in terms of skinfold thickness, they profited less regarding BMI. Without specific physical activity interventions, females might become obese during adolescence, a time of life when males are usually more physically active (e.g., playing soccer), and hence, specific interventions readily tailored to females might need to kick in later.

A Cochrane review revealed that school-based physical activity programmes are generally effective in increasing physical activity and physical fitness in children and adolescents aged 6–18 years [27]. Furthermore, physical activity programmes may also be useful to improve systolic blood pressure and heart rate [28,29]. However, the authors concluded that the magnitude of the effect is small. Moreover, while the review included 44 studies, all of them were carried out in Europe, North America and the People’s Republic of China, which precludes generalisation to an African context. Indeed, there is a paucity of research conducted among school-aged children in LMICs, including South Africa. Yet, setting-specific insights regarding the effectiveness of school-based initiatives are necessary to promote physical activity, which in turn might improve children’s health and wellbeing. We are aware of relevant previous research done in South Africa, specifically the HealthKick study [30]. The HealthKick study was conducted in primary schools in low-income settings, but the prime focus was on healthy nutrition. A common observation in LMICs is the paucity and often low quality of sport and recreation facilities, coupled with a lack of qualified teachers offering physical education classes. Absence or only irregular physical education complicate the promotion of age-appropriate physical activity, particularly in disadvantaged schools [31].

Previous studies in South Africa focusing on school-based interventions promoting a healthy lifestyle have shown disappointing results and the implementation of successful projects is even more difficult in children with lower SES, where the benefit could potentially be largest [27]. Infectious diseases and lack of physical activity among school-aged children and adolescents are important factors that influence growth. Furthermore, it adds to the complexity that anthropometry and physical fitness are impacted by a wide range of environmental and behavioural factors, including infectious diseases and declining physical activity in adolescence. Of note, several studies have highlighted a possible compensation of supervised physical activities in the school against spontaneous physical activity outside of school hours [32,33]. The findings of these studies support the hypothesis that more activity in school at one time is compensated for by less activity during leisure time. However, efforts should be made to increase the overall activity levels in youth because primary prevention against overweight during this time of life is essential. Previous investigations of physical activity patterns of primary schoolchildren in disadvantaged neighbourhoods in South Africa have confirmed insufficient physical activity levels [34]. For example, Kimani-Murage and colleagues reported that in a low-income South African setting, the co-existence of early stunting and adolescent obesity in females is a result of increasing physical inactivity [35]. This combination was particularly prevalent among black females, who showed the highest rates of physical inactivity [36]. As physical inactivity during childhood can lead to poor health outcomes in adulthood, promotion of physical activity among school-aged children is pivotal to prevent obesity-related conditions and related increased morbidity later in life [37]. The effect sizes of our physical activity intervention with respect to BMIZ and skinfold thickness raises hope that high intensity physical activity programmes in schools with a manageable duration can be used to combat excessive weight gain during adolescence. Additionally, in the face of weak healthcare systems, infectious diseases that are intimately connected with poverty are still widespread in disadvantaged South African schools [30]. Such dual burdens constitute a major challenge for the healthcare system, and infections might themselves have a negative impact on children’s nutritional status and cardiorespiratory fitness [38,39].

Compared to results from recent studies in Switzerland, the cardiorespiratory fitness of the South African study cohort was slightly below average. Swiss children achieved the same cardiorespiratory fitness level at seven years of age, three years earlier than their South African peers [40,41]. In contrast, Lang et al. reported slightly better reference values in South African than Swiss children [42]. Comparing our reference values with data obtained from a systematic literature review that included 50 countries, our values were slightly higher [43]. Also, Yap et al. reported slightly lower VO_2_ max results from south-west Yunnan province in the People’s Republic of China, namely 45.6 mL × kg^−1^ × min^−1^ for 9- to 12-year-old males and 44.7 mL × kg^−1^ × min^−1^ for females of the same age [39]. The high prevalence of chronic infections going hand-in-hand with fatigue and anaemia may contribute in part to the lower level of physical activity and complicate the performing of physical activity interventions. Furthermore, result modification of skinfold thickness by anaemia suggests that anaemia should be managed before accomplishing physical activity interventions, or else children might feel too tired to benefit from specific physical activity interventions.

Implementing a multidimensional physical activity intervention in schools in poor neighbourhoods was a challenging endeavour. Indeed, marginalised inhabitants of Port Elizabeth are still affected by the former Apartheid system [44]. A novelty of our study was the focus on health indicators of young primary schoolchildren in disadvantaged neighbourhoods, in communities with low SES and a high risk of inhabitants to develop obesity.

We were able to create, implement and measure the effect of a multidimensional physical activity intervention programme within an existing and productive research partnership, ensuring local relevance and acceptability and thus promoting sustainability. This intervention was perceived positively by the teachers (evaluation result: 4.2 on a 5-point Likert scale; 1 being poor and 5 excellent) and teachers and other education authorities expressed an interest to pursue the physical activity programme in the long term. The schoolchildren especially liked the weekly dance-to-music classes, perhaps as children are intrinsically motivated to move to music. The implementation of the intervention resulted in favourable effects on body composition, one of the major public health targets in LMICs [9]. Moreover, while school-based physical activity promotion has a long tradition in Western countries, the study presented here is among the few physical activity interventions implemented in an African context. Importantly, only few physical activity intervention programmes proved to be effective with regard to lowering the increase of BMI and body fat, the latter measured as thickness of skinfolds, in school-aged children [45]. We therefore assume that a multidimensional intervention might be particularly beneficial.

Our study has several limitations. First, the relevant motivation of teachers [46] to facilitate physical education classes was variable and influenced by unequal class sizes, the lack of formal qualifications for physical education and the fact that physical education lessons are generally not taught on a regular basis. Second, to achieve higher effects, an extended duration of the intervention, coupled with more extensive involvement of the school communities, school volunteers and parents/guardians may be needed, which might require changes in policies [27]. Third, no objective measurements of physical activity using accelerometers were pursued, which would have presented an objective and fine-grained picture [47]. Fourth, the sample size was relatively small and the intervention was only implemented in three out of eight schools for two 10-week periods because of logistic reasons. Fifth, concealment and blinding were not possible in our study design. Cluster-randomised trials have a greater potential for bias than studies that allow concealment and blinding. Despite these limitations, the study provides evidence for the feasibility of designing, implementing and evaluating a multidimensional physical activity intervention in a low-resource setting [11,48].

Physical education in schools needs to consider local resources, conditions and preferences, while adhering to minimal standards to be effective. Our intervention was developed by physical education specialists in consultation with local stakeholders. Careful adaptation will be necessary before the intervention can be implemented and scaled up in other settings. Moreover, a longer duration, preferably covering the entire school year and expansion to include all school grades is imperative. Lastly, the promotion of extra-curricular physical activity and healthy nutrition should become an integral part of the programme. New donors have agreed to fund such an expanded intervention in South Africa and two additional countries in East and West Africa, and the research-cum-action has started.

## 5. Conclusions

Our cluster-randomised controlled trial provides evidence that a well-designed multidimensional physical activity programme can lower the increase in BMI and thickness of skinfolds in school-aged children from disadvantaged communities in Port Elizabeth, South Africa, but no significant effects on cardiorespiratory fitness were observed. To increase effectiveness and sustainability of the results, the intervention should be extended to cover the entire school term and adapted to additional age groups. As overweight is caused by diverse lifestyle behaviours, there is a need for longitudinal monitoring, with age, gender and school-grade specific assessments. Furthermore, dissemination of this multidimensional, school-based physical activity programme in South Africa and elsewhere in sub-Saharan Africa may help to reduce risk factors for the development of chronic diseases among socially deprived children.

## Figures and Tables

**Figure 1 ijerph-16-00232-f001:**
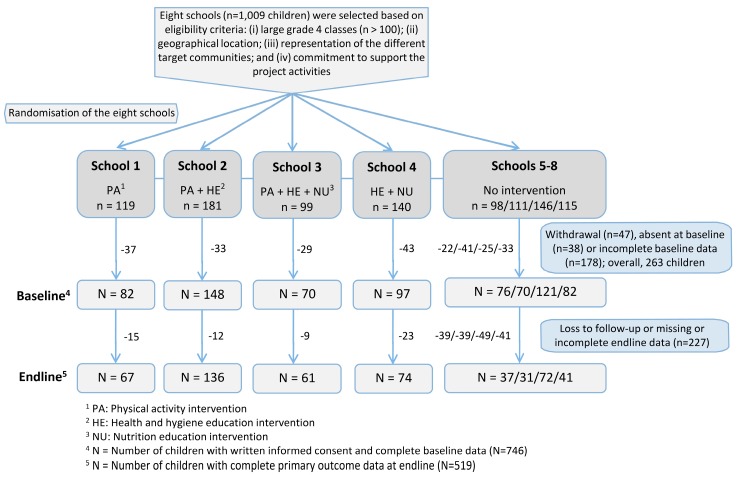
Selection of intervention sites (schools) for the ’Disease, Activity and Schoolchildren’s Health‘ (DASH) study, including flow chart of study participants with detailed information on all intervention arms from the randomisation of schools to the endline assessment of schoolchildren, Port Elizabeth, South Africa, 2015 and 2016.

**Figure 2 ijerph-16-00232-f002:**
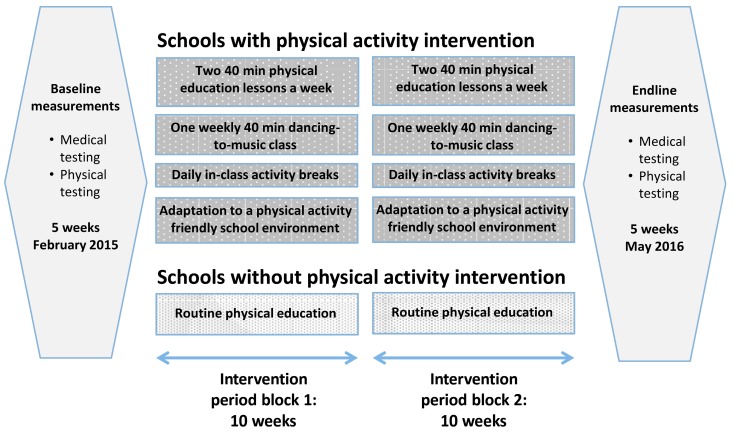
Timetable and content of the children’s assessments and the multidimensional physical activity intervention programme, Port Elizabeth, South Africa in 2015 and 2016. The total duration of physical activity for schools with physical activity intervention was 55 h, compared to 15 h for schools without physical activity intervention.

**Figure 3 ijerph-16-00232-f003:**
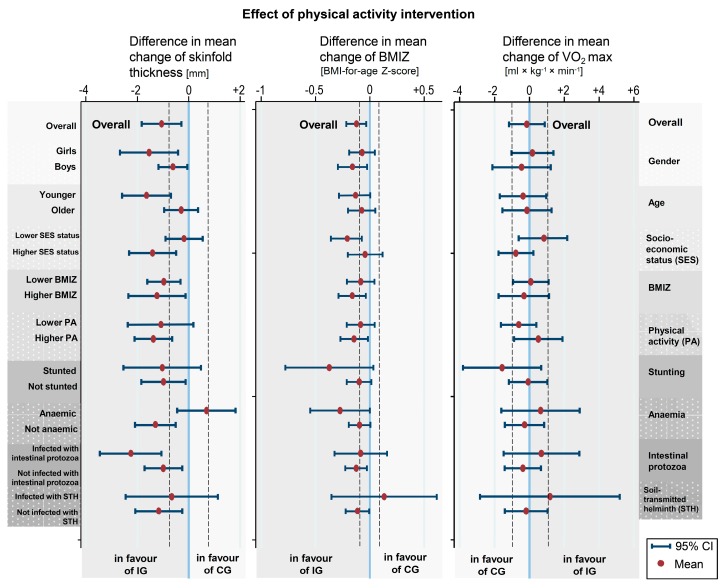
Estimated intervention effects on VO_2_ max, sex- and age-adjusted Z-score of body mass index (BMI) and thickness of skinfolds in different strata of children from baseline (February 2015) to 16-month endline (May 2016). Intervention effects were defined as differences in the mean longitudinal changes of the respective outcomes associated with the physical activity intervention.

**Table 1 ijerph-16-00232-t001:** Baseline characteristics of 746 children from Port Elizabeth, South Africa, in February 2015.

	Total (*n* = 746)	Schools with Physical Activity Intervention (*n* = 300)	Schools without Physical Activity Intervention (*n* = 446)
**In numbers (percentages)**			
Girls	372 (49.9)	150 (50.0)	222 (49.8)
Overweight ^a^	102 (13.7)	38 (12.7)	64 (14.3)
Obese ^b^	39 (5.3)	15 (5.0)	24 (5.4)
Stunted ^c^	86 (11.5)	46 (15.3)	40 (9.0)
Anaemic ^d^	138 (18.5)	67 (22.3)	71 (15.9)
Infected with intestinal protozoa ^e^	120 (16.1)	61 (20.3)	59 (13.2)
Infected with soil-transmitted helminths (STHs) ^f^	235 (31.5)	132 (44.0)	103 (23.1)
**In means (SD)**			
Age in years	10.0 (0.9)	10.1 (0.9)	9.9 (1.0)
Height in cm	133.3 (7.1)	132.6 (7.1)	133.8 (7.0)
Skinfolds in mm	9.0 (4.5)	9.0 (4.5)	9.0 (4.4)
Shuttle run in laps	36.3 (17.3)	35.6 (17.0)	36.8 (17.4)
VO_2_max ^g^ in mL × kg^−1^ × min^−1^	46.1 (4.3)	45.8 (4.1)	46.3 (4.3)
Overall SES index ^h^	0.0 (2.8)	−0.1 (2.7)	0.0 (2.9)
Poorest	−4.8 (2.3)	−4.5 (2.5)	−5.0 (2.1)
Second quintile	−0.3 (0.6)	−0.2 (0.6)	−0.4 (0.6)
Less poor	1.0 (0.2)	1.0 (0.2)	1.0 (0.2)
Fourth quintile	1.7 (0.2)	1.7 (0.2)	1.7 (0.2)
Least poor	2.3 (0.2)	2.3 (0.2)	2.3 (0.2)
Score of self-reported physical activity	8.4 (3.8)	9.1 (3.7)	7.8 (3.8)

^a^ Overweight: >+1 SD (equivalent to BMI 25 kg/m^2^ at 19 years); ^b^ Obesity: >+2 SD (equivalent to BMI 30 kg/m^2^ at 19 years); ^c^ Stunting is defined as height-for-age Z-score (HAZ) score <−2; ^d^ Anaemic is defined as haemoglobin concentration in blood ≤11.4 g dL^−1^; ^e^ Infected with one or two intestinal parasite species (*Cryptosporidium* spp. and/or *Giardia intestinalis*); ^f^ Infected with one or two soil-transmitted helminth (STH) species (*A. lumbricoides* and/or *T. trichiura*; no hookworm infections were diagnosed); ^g^ Using age-adjusted test protocol from Léger et al. [20]; ^h^ Socioeconomic status (SES) is based on self-reported household characteristics and assets, and calculated based on factor scores of principal component analysis (PCA).

**Table 2 ijerph-16-00232-t002:** Cardiorespiratory fitness and obesity outcome measures among children from Port Elizabeth, South Africa, at baseline (February 2015) and after a multidimensional physical activity intervention at the 16-month endline survey (May 2016). Values are unadjusted means (standard deviations) unless specified otherwise and estimated effects of the physical activity intervention on the mean changes in the respective outcome measures between baseline and endline, adjusted for the respective baseline value of sex, age, HAZ, SES index, Hb, soil-transmitted helminth (*A. lumbricoides* and *T. trichiura*) and intestinal protozoa (*Cryptosporidium* spp. and *G. intestinalis*) infection.

Variables	Schools with Physical Activity Intervention		Schools without Physical Activity Intervention		Intervention Effect ^a^		
	Baseline	Endline	Baseline	Endline	Estimate ^b^ (95% CI)	*p*-Value	ICC ^c^
	(*n* = 300)	(*n* = 264)	(*n* = 446)	(*n* = 255)			
Cardiorespiratory fitness							
Shuttle run (laps)	35.6	34.5	36.8	35.3	−0.56 (−4.67 to 3.56)	0.79	0.04
	(17.0)	(17.9)	(17.4)	(18.7)			
VO_2_max ^d^ (mL × kg^−1^ × min^−1^)	45.8	43.5	46.3	44	−0.14 (−1.17 to 0.88)	0.78	0.03
	(4.1)	(4.7)	(4.3)	(4.8)			
**Obesity**							
BMIZ ^e^	−0.1	−0.1	0	0.2	−0.17 (−0.24 to −0.09)	<0.001	<0.01
	(1.2)	(1.3)	(1.2)	(1.3)			
Skinfolds ^f^ (mm)	9.0	9.6	9	10.1	−1.06 (−1.83 to −0.29)	0.007	0.02
	(4.5)	(4.6)	(4.4)	(5.9)			
Mean of self-reported physical activity ^g^	9.1	9.0	7.8	9.9	−1.08 (−2.36 to 0.18)	0.09	0.04
	(3.7)	(3.1)	(3.8)	(3.4)			

^a^ Schoolchildren from the intervention group accomplished a multidimensional physical activity intervention programme between baseline and endline, as described in Figure 2; ^b^ Estimate of the physical activity intervention effect on the change in the respective outcome measure from baseline to endline, with 95%-confidence interval, P-value and ICC. The underlying linear mixed models included binary factor variables for all three intervention programmes (i.e., the physical activity intervention, the health education and the nutrition education programme) along with the baseline values of age, sex, height-for-age Z-score (HAZ), haemoglobin, socioeconomic (SES) index, protozoa- and soil-transmitted helminth (STH) infection status and random effects for classes and schools; ^c^ Proportion of unexplained variance attributable to clustering within schools and classes (intraclass correlation coefficient; ICC); ^d^ Using age-adjusted test protocol from Léger et al. [20]; ^e^ Sex-adjusted BMI-for-age Z-score (BMIZ); ^f^ Average of six measurements (triceps and subscapular three times each); ^g^ Score generated based on self-reported physical activity in the personal free time over the past 7 days and intense exercises outside structured school hours (range: from 1 to 14; 14 being the most active).

**Table 3 ijerph-16-00232-t003:** Stunting, anaemia and intestinal parasite infections among children from Port Elizabeth, South Africa, at baseline (February 2015) and the 16-month endline survey (May 2016) for schools with and without physical activity intervention. Values are numbers (percentages) unless specified otherwise.

Binary Variables	Schools with Physical Activity Intervention		Schools without Physical Activity Intervention		Intervention Effect ^a^		
	Baseline	Endline	Baseline	Endline	Odds Ratio ^b^	(95% CI)	*p*-Value
	(*n* = 300)	(*n* = 264)	(*n* = 446)	(*n* = 255)			
Stunted ^c^	46 (15.3)	47 (17.8)	40 (9.0)	48 (12.8)	New occurrence	0.68 (0.23 to 2.07)	0.50
					Re-occurrence	0.68 (0.12 to 3.96)	0.67
					Prevalence	0.76 (0.09 to 6.53)	0.81
Anaemic ^d^	67 (22.3)	37 (14.0)	71 (15.9)	50 (13.4)	New occurrence	1.36 (0.53 to 3.49)	0.52
					Re-occurrence	0.82 (0.25 to 2.62)	0.73
					Prevalence	0.93 (0.38 to 2.30)	0.87
Infected with soil-transmitted helminths (STHs) ^f^	132 (44.0)	108 (40.9)	103 (23.1)	51 (13.6)	New occurrence	2.33 (0.03 to 186.00) ^g^	0.71
					Re-occurrence	3.44 (0.04 to 298.70) ^g^	0.59
					Prevalence	1.92 (0.47 to 7.80)	0.36
Infected with intestinal protozoa ^e^	61 (20.3)	54 (20.5)	59 (13.2)	38 (10.2)	New occurrence	1.37 (0.65 to 2.90)	0.41
					Re-occurrence	1.55 (0.58 to 4.17)	0.38
					Prevalence	1.23 (0.47 to 3.22)	0.68

^a^ Schoolchildren from the intervention group accomplished a multidimensional physical activity intervention programme between baseline and endline, as described in Figure 2; adjustment was also made for two health and nutrition education programmes conducted at some of the schools in either group; ^b^ Re-occurrence (proportion of lasting cases) and new occurrence (proportion of new cases): adjusted OR of the respective outcome at endline between intervention and control groups among children with respectively without the outcome at baseline in a mixed logistic regression model with random intercepts at the unit of school and class. Prevalence (proportion of cases): OR of the interaction term between follow-up period and intervention in a mixed logistic regression model with random intercepts at the unit of class and child. Variables additionally adjusted for in these models were age, sex, height-for-age Z-score (HAZ), haemoglobin, SES index, intestinal protozoa and soil-transmitted helminth (STH) infection status at baseline; ^c^ Stunting is defined as HAZ score <−2; ^d^ Anaemic is defined as haemoglobin concentration in blood ≤11.4 g dL^−1^; ^e^ Infected with one or two intestinal parasite species (*Cryptosporidium* spp. and *Giardia intestinalis*); ^f^ Infected with one or two soil-transmitted helminth species (*A. lumbricoides* and *T. trichiura*; no hookworm infections were diagnosed); ^g^ The wide 95% CIs reflect strong heterogeneity between schools, with the vast majority of STH-infected children occurring in only two of the eight schools.

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
