# Peer review of "Effect of a Multidimensional Physical Activity Intervention on Body Mass Index, Skinfolds and Fitness in South African Children: Results from a Cluster-Randomised Controlled Trial"

_ijerph, 2019, doi:10.3390/ijerph16020232_

Round 1

Reviewer 1 Report

In relation to the manuscript entitled "Effect of a multidimensional physical activity intervention on 1 body mass index, skinfolds and fitness in South African 2 children: Results from a cluster randomized controlled trial", the authors present an intervention of interest for the journal line , although modifications are required before it can be accepted:

- The article must adapt to the format of the journal, using the downloadable template that can be found on the platform.

- The abstract should be divided into the sections established by the regulations of the journal.

- The introduction is correct, although a little short. It is recommended that the authors extend it, including a paragraph to look at the benefits of physical activity at the multidimensional level.

- The authors must include the research question at the end of the introduction and the hypothesis of their study (before the objectives).

- Include the section "Ethics statement" after the "Procedure" section.

- The sample and study population is correct. It is not necessary to include the geographic coordinates, so the authors must eliminate them.

- Please, indicate the average age and SD of the subjects.

- The analysis of the data and results of the study are correct. Also, the tables are self-explanatory and well designed.

- The discussion is correct and uses current references. Here are some references that could help improve it:

or López, G.F .; Nicolás, J .; Díaz, A. (2018). Effects of a vigorous physical activity program on blood pressure and heart rate of schoolchildren aged 10-11 years. Journal of Sport and Health Research, 10 (1): 13-24.

or Erturan-Ilker, G .; Yu, C .; Alemdaroğlu, U .; Köklü, Y. (2018). Basic psychological needs and self-determined motivation in PE to predict health-related fitness level. Journal of Sport and Health Research, 10 (1): 91-100.

- The conclusions are adequate. Please add the answer to the research question and the answer to your initial hypotheses.

- The authors should review the bibliographic references and add the DOI.

Author Response

Prof. Dr. David A. Sleet

Section Editor Children Health

Müller et al. “Effect of a multidimensional physical activity intervention on body mass index, skinfolds and fitness in South African children: Results from a cluster randomised controlled trial” (your reference no. IJERPH-404592) – point-by-point response

Basel and Port Elizabeth, 15 December 2018

Dear Prof. Sleet

We refer to your e-mail dated 8 December 2018 and thank you and the three external reviewers very much indeed for all your efforts with our piece and the opportunity to revise and resubmit. In the meantime, we carefully revised the aforementioned manuscript in light of the reviewers’ comments and suggestions. Below, please find our point-by-point response, clearly indicating how and where in the manuscript (line numbers) changes have been made.

As per your request, we are re-submitting the document electronically. Please note that the reviewers’ points are in italics, while our non-italicized responses follow each point in a grey shaded box. In the edited manuscript, we used the track changes function in MS Word.

We very much hope that our revised manuscript is deemed suitable for publication in the International Journal of Environmental Research and Public Health. Hence, we look forward to your further disposition.

Yours sincerely,

Ivan Müller (on behalf of all authors)

*********************************************************************************

Response #1:

Open Review

English language and style

( ) Extensive editing of English language and style required
( ) Moderate English changes required
( ) English language and style are fine/minor spell check required
(x) I don't feel qualified to judge about the English language and style

Yes

Can be   improved

Must be   improved

Not   applicable

Does the introduction   provide sufficient background and include all relevant references?

( )

(x)

( )

( )

Is the research design   appropriate?

(x)

( )

( )

( )

Are the methods   adequately described?

( )

(x)

( )

( )

Are the results clearly   presented?

( )

( )

( )

( )

Are the conclusions   supported by the results?

( )

(x)

( )

( )

Comments and Suggestions for Authors

In relation to the manuscript entitled "Effect of a multidimensional physical activity intervention on 1 body mass index, skinfolds and fitness in South African 2 children: Results from a cluster randomized controlled trial", the authors present an intervention of interest for the journal line , although modifications are required before it can be accepted:

- The article must adapt to the format of the journal, using the downloadable template that can be found on the platform.

Response #1: We thank Reviewer #1 for this comment. We moved the "Ethics statement" behind the "Procedure" section.

- The abstract should be divided into the sections established by the regulations of the journal.

Response #2: We followed the IJERPH guidelines for authors with regards to the abstract: “The abstract should be a single paragraph and should follow the style of structured abstracts, but without headings: 1) Background: Place the question addressed in a broad context and highlight the purpose of the study; 2) Methods: Describe briefly the main methods or treatments applied. Include any relevant preregistration numbers, and species and strains of any animals used. 3) Results: Summarize the article's main findings; and 4) Conclusion: Indicate the main conclusions or interpretations.” Hence, we believe that we have respected and followed the journal guidelines.

- The introduction is correct, although a little short. It is recommended that the authors extend it, including a paragraph to look at the benefits of physical activity at the multidimensional level.

Response #3: We have added a paragraph on the benefits of school-based physical activity programmes on fatness and fitness outcomes, as follows “A meta-analysis by Guerra and colleagues with data from 11 randomized trials suggests that, regardless of the potential benefits of physical activity in the school environment, school-based physical activity interventions did not have any significant effects on body mass index [6], while another non-randomized trial by Li et al. reported favourable effects [7]” (see revised manuscript, lines 65-69).

- The authors must include the research question at the end of the introduction and the hypothesis of their study (before the objectives).

Response #4: The hypothesis of our study is now clearly articulated before the objectives and at the end of the Introduction (see revised manuscript, lines 80-84).

- Include the section "Ethics statement" after the "Procedure" section.

Response #5: We moved the "Ethics statement" behind the "Procedure" section.

- The sample and study population is correct. It is not necessary to include the geographic coordinates, so the authors must eliminate them.

Response #6: We considered this point; yet, we would like to keep the geographical coordinates, as this is important information for future geostatistical analyses.

- Please, indicate the average age and SD of the subjects.

Response #7: The requested specific information has been inserted, as follows “…with mean age=11.2 years (standard deviation=0.9 years)…” (see revised manuscript, line 136).

- The analysis of the data and results of the study are correct. Also, the tables are self-explanatory and well designed.

Response #8: We thank Reviewer #1 for these generous comments.

- The discussion is correct and uses current references. Here are some references that could help improve it:

or López, G.F .; Nicolás, J .; Díaz, A. (2018). Effects of a vigorous physical activity program on blood pressure and heart rate of schoolchildren aged 10-11 years. Journal of Sport and Health Research, 10 (1): 13-24.

Response #9: The proposed reference is spot on, and hence, we have slightly amended the Discussion, as follows: “Furthermore, physical activity programmes may also be useful to improve systolic blood pressure and heart rate [28, 29]” (see revised manuscript, lines 400-401).

or Erturan-Ilker, G .; Yu, C .; Alemdaroğlu, U .; Köklü, Y. (2018). Basic psychological needs and self-determined motivation in PE to predict health-related fitness level. Journal of Sport and Health Research, 10 (1): 91-100.

Response #10: The proposed reference has been included in the Discussion (see revised manuscript, line 477).

- The conclusions are adequate. Please add the answer to the research question and the answer to your initial hypotheses.

Response #11: In our view, the conclusion does answer to our research question and underlying hypothesis.

- The authors should review the bibliographic references and add the DOI.

Response #12: We carefully revised the references and, whenever possible, we added URLs and/or DOIs.

Reviewer 2 Report

Review of the manuscript ijerph-404592 entitled “Effect of a multidimensional physical activity intervention on  body mass index, skinfolds and fitness in South African children: Results from a cluster randomised controlled trial”. In this paper authors examined whether participation in the DASH multidimensional physical activity intervention program would improve children’s cardiorespiratory fitness and counteract an excess increase of body mass index  (BMI) and skinfold thickness in school-aged children from disadvantaged communities in Port Elizabeth (South Africa). Although after the physical activity program only a lower increase in BMI and thickness of skinfolds was observed, the results obtained raises hope that high intensity physical activity programs in schools can be useful to prevent excessive weight gain during adolescence.

The work is well organized. Some minor English error are present in the text (e.g. line 348 focussing).

Some minor considerations are reported below:

Materials and Methods

Lines 99-101 – The classification in quintile should be better explained

Results

Table 1 – Some information reported in the caption of the table1 should be described in the results section in order to better explain the results obtained.

Author Response

Prof. Dr. David A. Sleet

Section Editor Children Health

Müller et al. “Effect of a multidimensional physical activity intervention on body mass index, skinfolds and fitness in South African children: Results from a cluster randomised controlled trial” (your reference no. IJERPH-404592) – point-by-point response

Basel and Port Elizabeth, 15 December 2018

Dear Prof. Sleet

We refer to your e-mail dated 8 December 2018 and thank you and the three external reviewers very much indeed for all your efforts with our piece and the opportunity to revise and resubmit. In the meantime, we carefully revised the aforementioned manuscript in light of the reviewers’ comments and suggestions. Below, please find our point-by-point response, clearly indicating how and where in the manuscript (line numbers) changes have been made.

As per your request, we are re-submitting the document electronically. Please note that the reviewers’ points are in italics, while our non-italicized responses follow each point in a grey shaded box. In the edited manuscript, we used the track changes function in MS Word.

We very much hope that our revised manuscript is deemed suitable for publication in the International Journal of Environmental Research and Public Health. Hence, we look forward to your further disposition.

Yours sincerely,

Ivan Müller (on behalf of all authors)

*********************************************************************************

Response #2:

Open Review

English language and style

( ) Extensive editing of English language and style required
( ) Moderate English changes required
(x) English language and style are fine/minor spell check required
( ) I don't feel qualified to judge about the English language and style

Yes

Can be   improved

Must be   improved

Not   applicable

Does the introduction   provide sufficient background and include all relevant references?

(x)

( )

( )

( )

Is the research design   appropriate?

(x)

( )

( )

( )

Are the methods   adequately described?

( )

(x)

( )

( )

Are the results clearly   presented?

( )

(x)

( )

( )

Are the conclusions   supported by the results?

(x)

( )

( )

( )

Comments and Suggestions for Authors

Review of the manuscript ijerph-404592 entitled “Effect of a multidimensional physical activity intervention on body mass index, skinfolds and fitness in South African children: Results from a cluster randomised controlled trial”. In this paper authors examined whether participation in the DASH multidimensional physical activity intervention program would improve children’s cardiorespiratory fitness and counteract an excess increase of body mass index (BMI) and skinfold thickness in school-aged children from disadvantaged communities in Port Elizabeth (South Africa). Although after the physical activity program only a lower increase in BMI and thickness of skinfolds was observed, the results obtained raises hope that high intensity physical activity programs in schools can be useful to prevent excessive weight gain during adolescence.

The work is well organized. Some minor English error are present in the text (e.g. line 348 focussing).

Response #13: We thank Reviewer #2 for reading our piece so carefully and for pointing out some spelling mistakes. These have now been corrected (see revised manuscript, line 414).

Some minor considerations are reported below:

Materials and Methods

Lines 99-101 – The classification in quintile should be better explained

Response #14: We added the following information: “South Africa’s public schools are classified into five groups, with quintile five standing for the least poor and quintile one standing for the poorest. The quintiles are determined through the national poverty table, prepared by the treasury. Areas are being ranked on the basis of income levels, dependency ratios and literacy rates in the area. The quintile ranking of a school determines the no-fee status of the school and also the amount of money that a school receives, with the poorest schools receiving the greatest per-learner allocation [15]” (see revised manuscript, lines 113-119).

Results

Table 1 – Some information reported in the caption of the table1 should be described in the results section in order to better explain the results obtained.

Response #15: Adjustments have been made and the corresponding sentences now read as follows: “Table 1 summarises the baseline characteristics of the study participants, stratified by schools with physical activity intervention and schools without physical activity intervention. No significant differences in primary outcome measures, such as obesity, skinfolds and cardiorespiratory fitness at baseline were detected, when comparing schools with and without physical activity intervention (all P>0.05)” (see revised manuscript, lines 317-322).

Reviewer 3 Report

The present study determined the effect of a multidimensional school-based physical activity intervention on fatness and fitness in South-African 4th graders. There are some issues that need to be clarified to be able to evaluate the study’s validity. Moreover, I have several comments to improve the paper. 1) There are no information on the available evidence of school-based physical activity interventions’ effect on fatness and fitness outcomes in the introduction, while there are a lot of evidence, including systematic reviews and meta-analyses, available in this field. Please include what is known in this field up front, although it is discussed later. 2) Recruitment: Please elaborate on the generalizability of the results, given that only selected schools were included. These seems to be the one best fit for the intervention. Moreover, please provide a flow diagram according to the CONSORT statement, detailing both/all intervention arms separately and clearly stating the numbers of schools and children included, excluded and analyzed. This might also help me to understand the next issue (3). 3) Please describe the design and different interventions in more details. Did the same children take part in several intervention components? If so is the case, how is it possible to separate the effects of the different components? Based on the findings, showing favourable effects for fatness but not fitness (which is opposite the findings by Resaland, G. K., et al. (2017). "The effect of a two-year school-based daily physical activity intervention on a clustered CVD risk factor score-The Sogndal school-intervention study." Scandinavian Journal of Medicine & Science in Sports. – a study that should be included in the discussion of this issue); could the effect be ascribed to the nutrition intervention and not the physical activity intervention? 4) Intervention/Figure 2. Please provide exactly (as far as possible) the total duration of PA for the intervention and control groups. 5) I find the statistical analyses very unclear. Please describe clearly how you tested the intervention effects; I suppose you tested a group*time interaction using a linear mixed model for the main outcomes of the study; fatness and fitness variables (which are continuous). I am confused by the detailed description of many covariates and/or moderating variables and description of several logistic regression models. The main outcomes are continuous and should be rather straight-forward to test in an RCT design. How can the OR for an interaction term (line 237) be used as the effect estimate of the physical activity intervention, when the effects are reported in (BMI) z–score units and (skinfold) cm? 6) Line 376, when discussion 20 m shuttle run performance: I suggest using international reference values rather than comparing to Swiss studies alone (Tomkinson, G. R., et al. (2017). "International normative 20 m shuttle run values from 1 142 026 children and youth representing 50 countries." British Journal of Sports Medicine 51(21): 1545-).

Author Response

********************************************************************************

Response #3:

Open Review

English language and style

( ) Extensive editing of English language and style required
( ) Moderate English changes required
(x) English language and style are fine/minor spell check required
( ) I don't feel qualified to judge about the English language and style

Yes

Can be   improved

Must be   improved

Not   applicable

Does the introduction   provide sufficient background and include all relevant references?

( )

( )

(x)

( )

Is the research design   appropriate?

( )

(x)

( )

( )

Are the methods   adequately described?

( )

( )

(x)

( )

Are the results clearly   presented?

( )

(x)

( )

( )

Are the conclusions   supported by the results?

( )

( )

(x)

( )

Comments and Suggestions for Authors

The present study determined the effect of a multidimensional school-based physical activity intervention on fatness and fitness in South-African 4th graders. There are some issues that need to be clarified to be able to evaluate the study’s validity. Moreover, I have several comments to improve the paper.

1) There are no information on the available evidence of school-based physical activity interventions’ effect on fatness and fitness outcomes in the introduction, while there are a lot of evidence, including systematic reviews and meta-analyses, available in this field. Please include what is known in this field up front, although it is discussed later.

Response #16: As already explained in our response #3, we added a paragraph on the benefits of school-based physical activity programmes on fatness and fitness outcomes.

2) Recruitment: Please elaborate on the generalizability of the results, given that only selected schools were included. These seems to be the one best fit for the intervention. Moreover, please provide a flow diagram according to the CONSORT statement, detailing both/all intervention arms separately and clearly stating the numbers of schools and children included, excluded and analyzed. This might also help me to understand the next issue (3).

Response #17: To address external validity, let us clarify our procedures once more (in the paper from lines 113-130): All quintile 3 primary schools in the Port Elizabeth district (n = 103) were invited to participate in the study. Of note, public schools in South Africa are classified into five groups, with quintile five standing for the least poor and quintile one standing for the poorest. The quintiles are determined through the national poverty table, prepared by the treasury. Areas are being ranked on the basis on income levels, dependency ratios and literacy rates in the area. The quintile ranking of a school determines the no-fee status of the school and also the amount of money that a school receives, with the poorest schools receiving the greatest per-learner allocation (Hall et al. (2009): Addressing quality through school fees and school funding). From the 103 quintile three schools, only 25 schools expressed an interest that was witnessed by a written response. These 25 schools were invited to an information sharing meeting, and 15 schools were represented, whereas 10 schools declined to attend the meeting (please see revised Figure 1). The final eight school were selected based on (i) sufficiently large grade 4 classes (n>100); (ii) geographical location of schools (here, we were mainly interested in schools that were sufficiently well spread out to avoid cross-contamination between intervention and control schools); (iii) representation of the various target communities: quintile 3 schools are situated in historically disadvantaged areas (note that in Port Elizabeth, there are areas known as “township areas”, inhabited predominantly by black African people and the “Northern areas”, inhabited by predominantly coloured (mixed-race) people; both these communities needed to be represented); and (iv) commitment to support the project activities. Of the remaining 15 schools, seven did not satisfy the criterion of having at least 100 children in grade 4. Due to financial constraints, logistics, and limited manpower, we were able to implement the physical activity intervention programme in only three out of the eight schools.

Moreover, a flow chart diagram according to the CONSORT statement regarding the physical activity intervention can be seen below (Figure 1) with final cohort (Table Supplementary Material S1).

Figure 1: Please find the Figure 1 in the attached word document "IJERPH-404592-Research Manuscript R1 (15.12.2018)".

Table Supplementary Material S1: Please find the Table Supplementary Material S1 in the attached word document "IJERPH-404592_Electronic Supplementary Material R1 (15.12.2018)".

For comprehensive understanding with regards to the number of schools and schoolchildren included in all the intervention arms, please find below two tables “Intervention schemes and sample sizes at baseline and endline across the eight schools”. Of note, (i) physical activity; (ii) health/hygiene education; and (iii) nutrition education intervention carried out in the final eight schools:

·      At the endline measurement, 519 children (264 with physical activity intervention and 255 without physical activity intervention) were analysed.

·      At the endline measurement, 519 children (271 with health/hygiene education and 248 without health/hygiene education) were analysed.

·      At the endline measurement, 519 children (135 with nutrition education intervention and 384 without nutrition education intervention) were analysed.

Intervention schemes and sample sizes at baseline across the eight schools

Study arm

Schools   with physical activity intervention

Schools   without physical
  activity intervention

Physical activity

School 1 (76   children1)

Physical activity and   health/hygiene education

School 2 (148   children1)

Physical activity, health/hygiene   education and nutrition education

School 3 (76   children1)

(No physical activity), health/hygiene   education and nutrition education

School 4 (97   children1)

(No physical activity)

Schools 5,   6, 7 and 8 (349 children1)

1 The number of included children at baseline with complete baseline data.

Intervention schemes and sample sizes at endline across the eight schools

Study arm

Schools   with physical activity intervention

Schools   without physical
  activity intervention

Physical activity

School 1   (67 children2)

Physical activity and   health/hygiene education

School 2   (136 children2)

Physical activity, health/hygiene   education and nutrition education

School 3   (61 children2)

(No physical activity), health/hygiene   education and nutrition education

School 4   (74 children2)

(No physical activity)

Schools 5,   6, 7 and 8 (181 children2)

2 The number of analysed children at endline with complete baseline data but incomplete endline data.

3) Please describe the design and different interventions in more details. Did the same children take part in several intervention components? If so is the case, how is it possible to separate the effects of the different components? Based on the findings, showing favourable effects for fatness but not fitness (which is opposite the findings by Resaland, G. K., et al. (2017). "The effect of a two-year school-based daily physical activity intervention on a clustered CVD risk factor score-The Sogndal school-intervention study." Scandinavian Journal of Medicine & Science in Sports. – a study that should be included in the discussion of this issue); could the effect be ascribed to the nutrition intervention and not the physical activity intervention?

Response #18: The effects of the different intervention components can be distinguished because they were applied in different combinations. In our analyses, each component was represented by an indicator variable taking the value 1 if the component was implemented in the respective school and the value 0 in schools where the component was not implemented. The proposed reference, Resaland et al. (2017), is spot on, and hence, was included in the Discussion (line 401).

4) Intervention/Figure 2. Please provide exactly (as far as possible) the total duration of PA for the intervention and control groups.

Response #19: As far as possible, the calculations of the total duration of PA for the intervention group (schools with physical activity intervention) was 55 hours, compared to 15 hours for schools without physical activity intervention. We have included the following information in the caption of Figure 2: “The total duration of physical activity for schools with physical activity intervention was 55 hours, compared to 15 hours for schools without physical activity intervention.”

5) I find the statistical analyses very unclear. Please describe clearly how you tested the intervention effects; I suppose you tested a group*time interaction using a linear mixed model for the main outcomes of the study; fatness and fitness variables (which are continuous). I am confused by the detailed description of many covariates and/or moderating variables and description of several logistic regression models. The main outcomes are continuous and should be rather straight-forward to test in an RCT design. How can the OR for an interaction term (line 237) be used as the effect estimate of the physical activity intervention, when the effects are reported in (BMI) z–score units and (skinfold) cm?

Response #20: We changed the description of the mixed regression models used, with the aim of more clearly distinguishing analyses of quantitative outcomes (changes in quantitative measures from baseline to follow-up) and binary outcomes (occurrence of symptoms or infections) (see revised manuscript, lines 249-297). We also tried to make it clear to the reader that the effects of three different interventions were estimated in parallel. However, in accordance with the focus of the present paper, only the estimated effects of the physical activity intervention are reported in the main text, while the estimated effects of the two other interventions are reported in the on-line supplement. Regression models for changes in quantitative outcomes or for the new occurrence or re-occurrence of symptoms or infections at follow-up include only one outcome per child and therefore contain random intercepts of schools and classes but not of children. In contrast, models for prevalence include outcomes at baseline and follow-up as repeated measures and therefore additionally include a random intercept for each child. Moreover, models for prevalence also include an indicator variable for follow-up observations (period variable) and interaction terms between the three intervention indicator variables and the period variable. Results for quantitative outcomes are reported as adjusted mean changes and those for binary outcomes as adjusted odds ratios.

6) Line 376, when discussion 20 m shuttle run performance: I suggest using international reference values rather than comparing to Swiss studies alone (Tomkinson, G. R., et al. (2017). "International normative 20 m shuttle run values from 1 142 026 children and youth representing 50 countries." British Journal of Sports Medicine 51(21): 1545-).

Response #21: The following sentence has been added: “Comparing our reference values with data obtained from a systematic literature review that included 50 countries, our values were slightly higher [43]” (see revised manuscript, lines 446-447).

Round 2

Reviewer 3 Report

I thank the authors for their answers to my concerns and for clarifying several issues in their paper. I appreciate that the authors have provided more information about the design and the different intervention groups. Still, I strongly recommend a standard CONSORT flow-chart, including all intervention arms, be included in the paper. Please remove Figure 1 and include this flow chart as a new Figure 1.

Author Response

Prof. Dr. David A. Sleet

Section Editor Children Health

Müller et al. “Effect of a multidimensional physical activity intervention on body mass index, skinfolds and fitness in South African children: Results from a cluster randomised controlled trial” (your reference no. IJERPH-404592) – point-by-point response R2

Basel and Port Elizabeth, 23 December 2018

Dear Prof. Sleet

We refer to your e-mail dated 18 December 2018 and thank you and an external reviewer very much indeed for all your efforts with our piece and the opportunity to further revise and resubmit. As per Reviewer #3’s request, Figure 1 has been reworked and a standard CONSORT flow-chart, including all intervention arms, has been included. Below, please find our point-by-point response, clearly indicating how and where in the manuscript (line numbers) further changes have been made.

We are re-submitting the document electronically. Please note that the points offered for consideration by Reviewer #3 are in italics, while our non-italicized responses follow each point in a grey shaded box. In the edited manuscript, we used the track changes function in MS Word.

We very much hope that our further revised manuscript is deemed suitable for publication in the International Journal of Environmental Research and Public Health. Hence, we look forward to your further disposition.

Yours sincerely,

Ivan Müller (on behalf of all authors)

************************************************************************

Response #3:

Open Review

English language and style

( ) Extensive editing of English language and style required
( ) Moderate English changes required
(x) English language and style are fine/minor spell check required
( ) I don't feel qualified to judge about the English language and style

Yes

Can be   improved

Must be   improved

Not   applicable

Does the introduction provide sufficient background and include all   relevant references?

( )

(x)

( )

( )

Is the research design appropriate?

( )

(x)

( )

( )

Are the methods adequately described?

( )

(x)

( )

( )

Are the results clearly presented?

( )

(x)

( )

( )

Are the conclusions supported by the results?

( )

(x)

( )

( )

Comments and Suggestions for Authors

I thank the authors for their answers to my concerns and for clarifying several issues in their paper. I appreciate that the authors have provided more information about the design and the different intervention groups. Still, I strongly recommend a standard CONSORT flow-chart, including all intervention arms, be included in the paper. Please remove Figure 1 and include this flow chart as a new Figure 1.

Response #1: We thank Reviewer #3 for this comment. We have replaced the previous Figure 1 by a flow-chart that describes all five intervention arms from the randomisation of schools to the endline assessment of schoolchildren.

Round 3

Reviewer 3 Report

-